# Extensible Metaverse Implication for a Smart Tourism City

**Pannee Suanpang [1,*], Chawalin Niamsorn [2], Pattanaphong Pothipassa [3], Thinnagorn Chunhapataragul [1], Titiya Netwong [1]** and **Kittisak Jermsittiparsert [4,5,6,7,8]**

1. Faculty of Science & Technology, Suan Dusit University, Bangkok 10300, Thailand
2. Faculty of Management Science, Suan Dusit University, Bangkok 10300, Thailand
3. Faculty of Liberal Arts and Sciences, Sisaket Rajabhat University, Sisaket 33000, Thailand
4. Faculty of Education, University of City Island, Famagusta 9945, Cyprus
5. Faculty of Social and Political Sciences, Universitas Muhammadiyah Sinjai, Kabupaten Sinjai 92615, Sulawesi Selatan, Indonesia
6. Faculty of Social and Political Sciences, Universitas Muhammadiyah Makassar, Kota Makassar 90221, Sulawesi Selatan, Indonesia
7. Publication Research Institute and Community Service, Universitas Muhammadiyah Sidenreng Rappang, Sidenreng Rappang Regency 91651, South Sulawesi, Indonesia
8. Sekolah Tinggi Ilmu Administrasi Abdul Haris, Kota Makassar 90000, Sulawesi Selatan, Indonesia
* Correspondence: pannee_sua@dusit.ac.th; Tel.: +66-22445972

**Abstract:** The metaverse is an innovation that has created the recent phenomenon of new tourism experiences from a virtual reality of a smart tourism destination. However, the existing metaverse platform demonstrated that the technology is still difficult to develop, as the service provider did not disclose the internal mechanisms to developers, and it was a closed system, which could not use or share the user's data across platforms. The aim of this paper was to design and develop an open metaverse platform called the "extensible metaverse", which would allow new developers to independently develop the capabilities of the metaverse system. The acquisition of this new technology was conducted through requirements analysis, then the analysis and design of the new system architecture, followed by the implementation, and the evaluation of the system by the users. The results found that the extended metaverse was divided into three tiers that created labels, characters, and virtual objects. Furthermore, the linking tier combined the 3D elements, and the deployment tier compiled the results of the link to use all three parts by using the Blender program, Godot Engine, and PHP + WebGL as their respective key mechanisms. This system was tested in Suphan Buri province, Thailand, which was evaluated by 428 users. The results of the metaverse satisfaction, created tourism experience, and overall satisfaction of the variation of the satisfaction of using the metaverse were 86.0%, 79.7%, and 92.9%, respectively. The relative Chi-square ($\chi^2$/df) of 1.253 indicated that the model was suitable. The comparative fit index (CFI) was 0.984, the goodness-of-fit index (GFI) was 0.998, and the model based on the research hypothesis was consistent with the empirical data. The root mean square error of approximation (RMSEA) was 0.024. In conclusion, the extended metaverse is more flexible than other platforms and also creates the user's satisfaction and tourism experience in the smart destination to support sustainable tourism.

**Keywords:** extensible metaverse; smart tourism; tourism intention; virtual reality

## 1. Introduction

The impact of the COVID-19 pandemic has changed the world in all dimensions, including economic, social, technology, and way of life of people worldwide. In particular, the world has experienced technological disruption that would have much significance in the new normal in the post COVID-19 pandemic period [1–3]. With the rapid emergence of 'digital transformation' that comprises virtual commerce, online education, and social networking, this will stimulate further exploration of future technology and significantly increase the combination and innovations in various industries by using technology as a tool

for creating value and a business-competitive advantage [4,5]. The tourism industry, which is the world's major economic industry, has become a significant mechanism for driving global business. This industry was directly affected by the pandemic, consequently causing a lot of damage from the interruption by the virus [2,3,6]. Furthermore, in the tourism business sector, there has been an adjustment to support tourism in the following era by applying modern information technology to increase efficiency in tourism management [4]. The trend of tourism in the post COVID-19 era shows that tourists would pay more attention to a 'virtual tour' for creating a virtual experience before traveling to the destination. Simultaneously, the tourism sector has adapted itself by using information technology to help operate the business, which may be in the form of smart tourism [2,7]. The travel approaches of the pre-pandemic period demonstrated that the innovation of virtual tourism experiences was growing in popularity [7,8]; one of the streamlined technologies of virtual tourism is the 'metaverse'.

In the meantime, around July 2021, this new form of technology received every spotlight worldwide when Facebook CEO Mark Zuckerberg announced that the company had changed its name to "Meta" with a new logo that looked like an infinity symbol reflecting the future business of Facebook. The company focused on creating the "Metaverse" or "Virtual Reality-VR" by "Mark Zuckerberg" is estimated to be active with one billion "Metaverse" people in the next 10 years [9]. The transition of Facebook from the world's largest social platform service company to a virtual dual world like metaverse was considered a turning point that received attention from all circles, as everyone sees the metaverse as becoming the bridge that connects the new world together.

The metaverse-associated virtual experience has a perspective on the future of technology and its potential impact on the tourism industry [10]. Simultaneously, the tourism industry is enthusiastic in exploring the way of using the metaverse for creating virtual hospitality and tourism experiences, products, and services for sending value to their consumers [3,10]. Metaverse technology refers to a three-dimensional virtual space that creates a sense of presence; hence, it is attracting attention from the hospitality and tourism industry [8,10]. The metaverse has made both IT business and social media users excited all over the world. However, they will not be able to use metaverse technology soon because there are still many factors that must be prepared before heading into a new virtual world, especially the preparation of both equipment and systems that are to be supported and the need to be developed to be more advanced. This includes augmented reality (AR), 3D object simulation technology that would bring virtual objects to the real world by using display devices, such as glasses, monitors, and smartphone screens, and virtual reality (VR) technology that would simulate a place in a virtual world which would be separate from the real world. Users would interact with the simulated locations through a keyboard, mouse, gloves, shoes, etc. Therefore, the metaverse is certain to become an integral part of the tourism industry by changing tourism behavior and simulating situations in the virtual world before going to the real destination. However, the tourism stakeholders in the industry are puzzled by several questions, such as the following: How would the metaverse influence the future dynamics of the industry? What effects would the metaverse have on tourism customers' decision-making behavior and consumption patterns? What real-life experiences could be transferred to the metaverse [8]?

Thailand is rated among the world's top five post-pandemic destinations [11] due to having beautiful natural and historical resources, and high-value hospitality services [2,3,6,8,12]. The tourism industry of Thailand was the major economic earner that contributed 5.56% of the gross domestic product (GDP) before the crisis of the COVID-19 pandemic [2,3,6]. This circumstance had a direct effect on the tourism industry and business sector that wanted to develop trust in their safety and build confidence among tourists. As a result, tourists consider their health and safety as major priorities, which were considerably increased, thus creating a new normal way [2,3,6,8,12]. Simultaneously, the tourists' behavior has changed, as they select short trips, drive to unseen tourist attractions and enjoy less crowded and more personal travel [12]. Moreover, tourists' behavior is changing by using

a smart tourism system to search for information for planning their travel and using the VR of the metaverse to simulate the tourism experience.

In addition, the global metaverse market is expected to reach USD 194.4 billion in 2022 and increase almost fourfold to USD 758.6 billion by 2026, according to Global Industry Analysts Inc. This has been enhanced by the increased interest in virtual spaces for work and leisure during the pandemic [13]. Even though Thailand's metaverse market is still in its early stages, it has been eagerly welcomed by both the public and private sectors. This is because Thailand is one of Southeast Asia's front runners in ecommerce and 5G technology with a digitally savvy population [13]. At the national level, the metaverse is compatible with the Thailand 4.0 initiative, in which technology, innovation, and creativity would be used to create a value-based economy. In the retail sector, large Thai corporations, such as Siam Piwat and the CP Group, have already incorporated the metaverse into their business strategies. Simultaneously, the metaverse was recently introduced in the 'Amazing Thailand Metaverse: Amazing Durian', the country's prototype project that used Web 3.0 technology and the application of digital assets to promote tourism to connect the virtual world and real world together. At the same time, the project aimed to help business operators as Thailand reopens and to also upgrade the technical capabilities in the tourism industry [11,12].

To come up with new technology of the metaverse across multiple platforms, and in reviewing the related research, the researchers found that the current metaverse system and technology are still in their infancy and lack many features [14], such as being difficult to customize or empowered and unable to run characters, scenes, or digital assets easily and interchangeably [15].

This research contributed to expanding the capabilities of the metaverse and showed its application to Thailand's tourism industry, as well as assessing the response of tourists using the new platform to prepare and manage tourism in the post-COVID-19 era in Suphan Buri province. The new knowledge contributed to the sustainability of smart cities, which could be used to create the impression of a virtual experience to explore the tourism program and services that would respond to the needs of tourists. These initial reasons motivated the researchers to come up with the concept of researching and developing a metaverse platform that would be easy to customize, be scalable, and could work across platforms with other metaverse systems. In addition to the metaverse causing an awakening in the business circle, its arrival has inspired researchers to see the potential and opportunities of parallel world technologies that could be the key to resolving the recession of the tourism business in Suphan Buri province. This would be a new smart tourism destination landmark for Thailand in order to increase tourist's satisfaction and to reduce the cost of tourism promotions in the tourism business sector therefore the extensible metaverse were implication for a smart tourism city.

Therefore, the aim of this paper was to develop a metaverse platform that would be easy to study and expand the capabilities and assessment of the metaverse users with the following contributions:

(1) A comprehensive practical framework for building applications using metaverse technology that would be easy to learn and allow developers to establish additional system capabilities.
(2) The component of the framework could work and demonstrate how it was integrated into the Godot game engine.
(3) Address some common technical challenges when integrating the components and provide such solutions in order to make the researchers' metaverse platform easily interoperable with other metaverse platforms.

Moreover, this paper manipulates the state of the art of the applying information technology discipline that impacts the research work as follows in Table 1.

**Table 1.** The impact of the research work.

| Area of Applying Information Technology | The Impact of the Research Work |
|---|---|
| AI | The extended metaverse system would be innovative, and a prototype to use AI to match travelers' interests with the features of a tour program. |
| Smart Tourism Destination (STD) | The extended metaverse system would create a personalized tourist program and simulate the physical appearance of the destination that could inspire and attract the tourist's interest. Moreover, it could plug into the smart city's application platform to promote smart tourism. |
| Smart City | The extended metaverse would be integrated with the smart tourism platfrom, which would assist in the sustainability of the smart city. |
| Metaverse | This research would help illustrate the imperfections of the metaverse and how to tailor it for each task. |
| 3D/Virtual Reality | This research would demonstrate how in-depth 3D and virtual reality knowledge would be useful for customizing and expanding the capabilities of metaverse technologies. |

The rest of the paper is organized as follows: Section 2 offers the literature review, whereas Section 3 discusses the proposed metaverse environment. In Section 4, the extensible metaverse for the tourism industry is presented. In Section 5, the conclusion and its discussions are illustrated.

## 2. Review Literature

### 2.1. Sustainable Smart City

A smart tourism destination (STD) is the new approach that utilizes an information technology innovation to support the hospitality and tourism industry in the destination [3,16,17]. Moreover, an STD applies technologies to create value, pleasure, and experiences for tourists and profit for the tourism business. Additionally, many works in the literature have characterized an STD as a destination that collects data to understand the tourists' needs and behavior so as to provide a better experience and services in real time through the platform [18,19]. After the COVID-19 pandemic, many countries have had a tourism policy for rebooting, rebuilding, and resilience so as to encourage tourists to come back; one of the strategies used an 'STD' to promote the smart tourism destination to enhance sustainability tourism development for achieving the Sustainable Development Goals (SDG) model [3,18,19].

Suphan Buri province (Figure 1) was selected as the research area for this study. This province is located in the Central Region of Thailand and has a long history. There is also evidence related to politics, administration, arts and culture, and religion, with most of the population engaged in agriculture. Suphan Buri province is only 100 km from Bangkok, and the city itself dates back to the ninth century. Moreover, the province was once an important border town during the period of the Ayutthaya Kingdom; consequently, many battles of important wars were waged in this province. There are many historical sites, in addition to a wide variety of natural attractions. including caves, waterfalls, bird parks, fish sanctuaries, and smart farming. Additionally, the province has a policy to promote itself to be a smart city by promoting smart tourism, which is one of the significant mechanisms of the driving force of the economy of this area [19].

## Suphanburi Province Study Areas

**Figure 1.** Suphan Buri province research areas.

### 2.2. Virtual Tourism

In today's era where technology and the digital environment play an increasingly important role in human life, in terms of the tourism business, the concept of virtual tourism has emerged as one of the types and forms of tourism, and some have said that "Virtual tourism has been an interesting category for at least two decades since the creation of the virtual world" [20].

Nevertheless, virtual travel is here to stay in the digital realm of the web for listening to audio, reading text, and watching visual information of the body and the world around us. There are also unrealistic trips that can take tourists to natural attractions, ancient sites, museums, etc., for users who want to visit these sites and do not have the opportunity to visit [20]. Today, humans have recognized that hypertext tourism on the internet is possible with a single click of a mouse to visit palaces, museums, and cultural and ancient sites. The virtual tourism system allows users to experience the history of ancient times. Additionally, their urban camera system is a tool that connects them to the digital realm, so they can be searched and photographed [20].

Therefore, there are many researchers' efforts for developing a more efficient virtual travel system. In order to improve and control the efficiency of the virtual tourism system, Juelu and Tingting [21] presented a cultural virtual tourism system designed by using virtual tourism technology for three-dimensional tourism. This was conducted by developing the characteristics of the cultural tourism resources using fusion technology for virtual three-dimensional quantitative tracking, modeling the overall structure of the virtual tourism system, and analyzing the functional modules of the virtual tour system and virtualization environment. In this regard, the development of cultural tourism resource characteristics in the system was created using software called Multigen Creator, which improved the ability to match the human–machine interaction of the virtual tourism system using quantitative fusion technology and tracking. Thus, the virtual travel system designed with human–computer interaction and more realistic simulation of travel scenes could be used to effectively promote the development of cultural tourism resources. Furthermore, the research by Guo and Wang [22] is a good example of how to start developing virtual tourism to support cultural and natural tourism. Moreover, this was in the same direction as our research team in bringing virtual tourism to help publicize and plan tourism for Suphan Buri province.

### 2.3. Metaverse in Tourism

The concept of the metaverse first occurred in 1992, in the cyberpunk genre science fiction novel titled "Snow Crash". The novel discussed a virtual space that everyone had in parallel to the truth. The word "metaverse" was a portmanteau of the prefixes "meta" (meaning beyond) and "universe". The term has often been used to describe the concept of future internet iterations, which consist of continuous three-dimensional virtual spaces shared and connected to the perceived virtual universe (see Figure 2) [23].

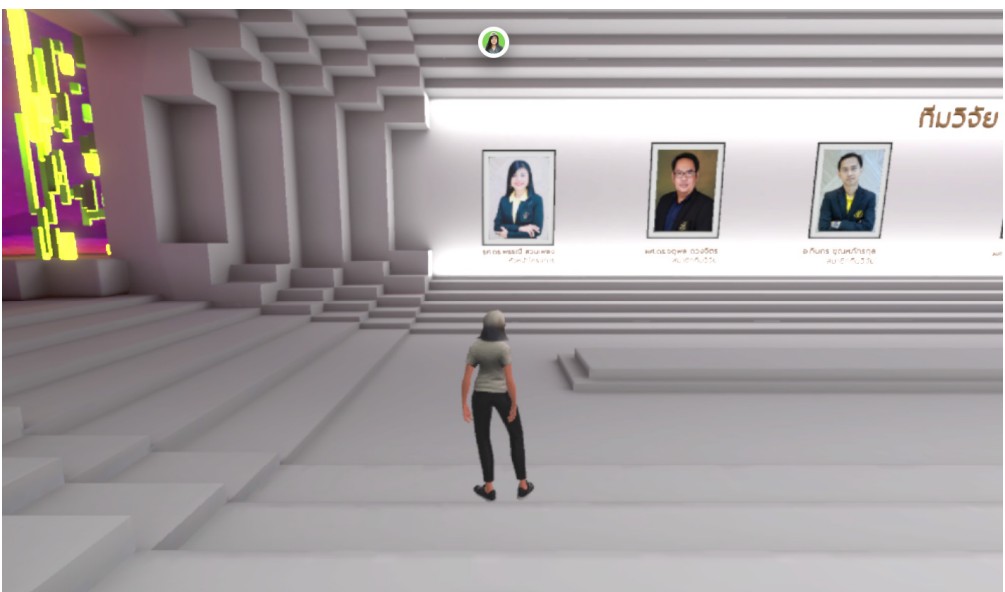

**Figure 2.** Example of metaverse [23].

In the research of Chand [23] and Sébastien et al. [24], three-dimensional technology enabled the creation of new information presentations by creating a virtual world, also known as "metaverses". The metaverses were created by combining three-dimensional models that could be viewed from any angle, from anywhere, at any time (spatial visualization). In these three-dimensional computational data, shapes could represent any object with a variable level of detail (LOD). Because metaverses offer visualizations closer to reality, they are easy to evaluate and understand [23,24].

Today, the global community's view of metaverse-related technologies began to change when Mark Zuckerberg officially announced that he would transform the popular social platform Facebook into the metaverse. Thus, the world's view of the metaverse, as put forward by Facebook CEO Mark Zuckerberg is as follows: Metaverse is a place where the real world and the digital world meet by the person. They will be able to create themselves as avatars that will look like the real individual or a fantasy character in any way. Avatars will be able to communicate, talk about anything, work, play, meet, go to a concert or even try on clothes without actually having to go to that place. Meta's Metaverse concept is like the "Virtual world" where everyone can go and perform activities in that world. By entering the world of Metaverse, it must be used through various devices. Activities that can be performed include watching movies, working, meetings, going to concerts, traveling to different places, or even trying on clothes without having to go to that place You can meet anyone, and you can do it without limitations in place, even when you are actually sitting at home. Access to the world of Metaverse can be granted in three ways: (i) Oculus glasses can be used to gain access to the world of VR, where you can create an avatar and live in the virtual world fully. (ii) Ray-Ban glasses or other glasses or devices with AR technology can be used which will give you moderate use. Think about playing Pokemon Go with Pokemon standing in real places. (iii) Finally, common devices, such as

computers or mobile phones, can be used, which will allow you to experience this virtual world the least.

The metaverse has been used for the hospitality and tourism industry for creating customer experiences, such as attending meetings, concerts, visiting museums and important tourist attractions, etc., which can be delivered in the metaverse system. This creates benefits by decreasing the tourist's carbon footprint while saving time, money, and other resources [8,25]. Moreover, the metaverse has changed the customer behavior to offer realistic hospitality and tourism experiences, especially as the necessary technology has emerged and user adoption has increased because of its functionality, that will have a significant role in the industry, enhancing marketing, customer relationships, communication, customer decision-making processes, and guest experiences. Additionally, the industry must exhibit ingenuity and innovation to create distinct and immersive experiences capable of engaging the customers' multiple senses [3,8,25].

Figure 3 illustrates the concept of the metaverse in the travel and tourism experience using a two-dimensional framework: (i) The first-dimensional degree of interactivity ranges from a low interactive experience (e.g., virtual flights and virtual concerts) to high interactivity (e.g., metaverse casino and virtual skydiving). (ii) The second dimension of the metaverse experience represents the types of motives of the customer, from hedonic motives that undertake a virtual experience for enjoyment and pleasure (e.g., virtual retailing, travel bookings, and education users) to a functional metaverse for facilitating hospitality and tourism services (e.g., digital twinning of a destination, hotel and resort and augmenting a physical experience) [8,26]. Finally, the metaverse offers significant opportunities to serve both tourism service providers and customers, while businesses use the metaverse for creating customer relationships, marketing, and support when they are planning trips. In addition, there would be opportunities to transform traditional hospitality and tourism experiences into virtual experiences. Customers would be able to use the metaverse for shopping, trying products and services before buying, enjoying concerts, or strolling through art galleries in the digital world [8,26]. Furthermore, Table 2 demonstrates a comparison of the metaverse and extended metaverse.

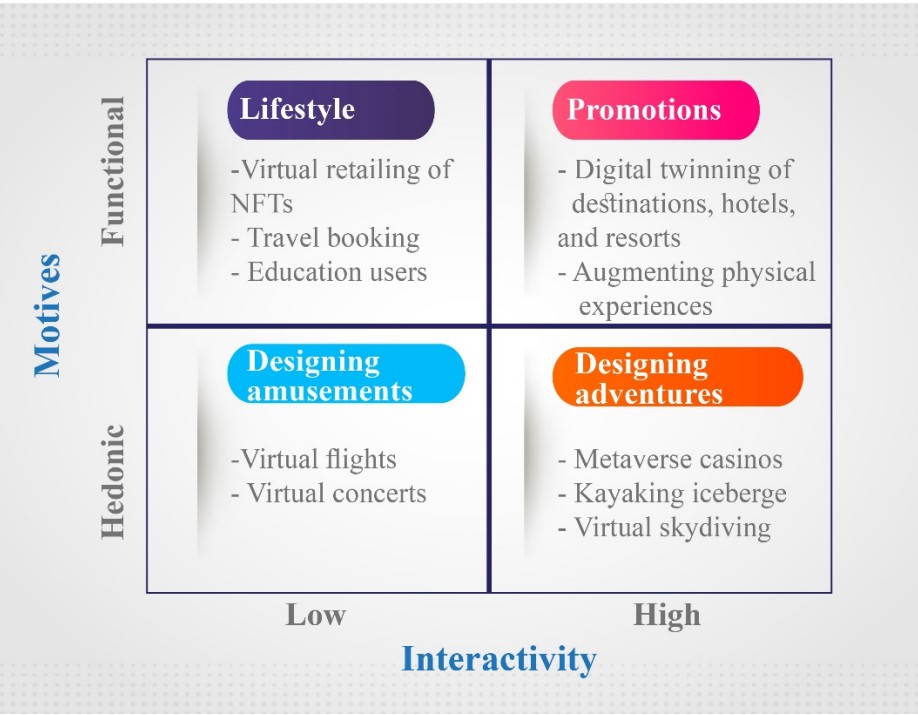

**Figure 3.** Metaverse experience [8].

**Table 2.** Comparison of the metaverse and extended metaverse.

| Abilities | Metaverse | Extended Metaverse |
|---|---|---|
| Flexibility to use | Medium | High |
| In terms of bringing back information | Low | High |
| Ease of customization | Low | High |
| Application aspects to artificial intelligence technology | Low | High |

### 2.4. Satisfaction of the Metaverse Experience

The metaverse user's experience could investigate the potential of the metaverse to modify various stages of the customer experience journey, such as the impact of using the metaverse journey on the consumers' outcome, such as overall customer engagement and customer satisfaction, and include the metaverse as an additional touchpoint in the customer experience journey with the hospitality and tourism industry [8]. Moreover, Gursoy et al. [8] suggested assessing the effectiveness of using VR or the metaverse for helping customers to prompt booking behavior and measured the value of using the metaverse for increasing the bookings of hospitality services, such as hotels, restaurants, museums, transportation, and other products and services. Many studies investigated tourist satisfaction and destination revisiting by developing behavioral models of ecotourism and examining the relationship between quality, satisfaction, and revisiting [16–18,22]. However, many studies looked at the tourists' expectations that would help to improve the quality of the services and overall satisfaction, which would influence their revisit as well as their recommendation of the destination to other potential visitors [16–18,27,28].

### 2.5. Blender

Blender is free software for three-dimensional computer graphics that can be used to create three-dimensional modeling, UV unfolding, texture, bone motion manipulation, water flow simulation, skin simulation, computer animation, rendering, particle, other computer simulations, edit videos and images via compositing systems, and can also be used to create three-dimensional applications. Blender works on many operating systems, including Microsoft Windows, Mac OS X, GNU/Linux, IRIX, Solaris, NetBSD, FreeBSD, and OpenBSD and is unofficially ported to systems such as BeOS, SkyOS, AmigaOS, MorphOS, and Pocket PC. Blender is on par with other high-end three-dimensional programs, such as Softimage|XSI, Cinema 4D, 3DS Max, Lightwave, and Maya with key features, such as falling object simulation, a collision between a fluid, and a cloth being blown by the wind. It includes a layered modifier for modeling, a high-quality animation handler, a node-based material handling and compositing system, and Python support for scripting. Blender requires OpenGL to operate. In 2007, Blender was the world's most installed three-dimensional animation software [26,27].

In the study [27], Blender was used in a novel approach that used three-dimensional modeling techniques to accurately and efficiently quantify the interstitial spaces in benthic marine habitats, including the area provided by the surface. The method was built on the analog methods routinely used on rocky shores and intertidal habitats, including roughness, terrain, fractions, and volume measurements, which were provided directly to the metaverse system. Thus, such measurements allowed the retrospective analysis of specific historical data on the composition of the species. It allowed users to quickly create and measure large quantities of samples at no additional cost, and when creating a three-dimensional model of the species, all methods would take less than 10 min to complete. This was as accurate as fractal analysis in determining the complexity of the structures on a rock but was more consistent, accurate, and distinguishable, whereas the approach by Stakoverflow [27], Bastian et al. [28], and Sadchatheeswaran [29] was consistent with our research team's method in terms of finding the ability to apply Blender to new tasks.

*2.6. WebGL*

The web graphics library (WebGL) is the use of a JavaScript programming language on a website, which has an application programming interface (API) to help in terms of rendering three dimensions and two dimensions, which must work under a web browser without the need to install additional plug-ins. WebGL is the standard by which all web browsers are able to process images and display various effects, which is part of the WebGL canvas and can be used in conjunction with HTML tags. WebGL is based on OpenGL ES 2.0 and provides an interface for three-dimensional graphics with the HTML5 canvas element and access to the document object model (DOM) interfaces.

Liu et al. [30] described the implementation of a three-dimensional representation of air pollutants in a network environment using open-source WebGL classes in conjunction with the JS Third Library and using kd-tree to locate the nearest data point to fix the pollutant's three-dimensional grid data as the source data, which were scattered. This made it easy to find information at the customer's destination. Binary files were also used to save and transfer the data to reduce the number of network transmissions and improve the speed of rendering. The results showed that the visual effects and rendering speeds were significantly improved on the web using the technology presented. In this study, the researchers presented NetV.js, an open-source and WebGL-based JavaScript library that supported the rapid rendering of large graph data (up to 50,000 nodes and one million edges) at an interactive frame rate with a commodity computer [30]. The results showed that the above libraries outperformed the existing toolkits (Sigma.js, D3.js, Cytoscape.js, and Stardust.js) in terms of performance.

*2.7. Godot Engine*

Godot is a game creation tool. This can be done in both two dimensions and three dimensions and is free to use. Its ability is considered to be able to compete with famous game engines, such as Unity or unreal engines. In addition, after finishing writing the game, the researchers' game could also operate on many platforms, including Windows, iOS, Android, or a web game, thus saving a lot of time in noting each platform down.

In addition, Matišák and Žáková [31,32] described a system that presented animations of mechatronic systems to students. It could operate in two modes—in an open loop to monitor the dynamic behavior of the system without any controls, and in a closed loop, considered a basic proportional–integral–derivative (PID) controller. The aim was to show students the effect of each component of the control, such as the effect of the P, I, and D segments. The researchers used three-dimensional animations that were available as web applications. The researchers then decided to use the increasingly popular AR. This type of simulation was available via smartphones and was based on Google's service for the AR platform built for AR experiences. Both types of visualizations were demonstrated in the tower copter models that the researchers had in our laboratory as well as in the actual factory. Using the presented educational environment gave the students their first experience with the system before using the actual equipment in the laboratory. This was conducted along with explaining how to use the three game engines, Godot, Unity and Unreal, as a web application. TAT News article [12] and Matišák and Žáková's [32] concept was consistent with our idea that creating virtual experiences and knowledge could be developed into the form of a web application.

*2.8. Extended Metaverse*

In the research, T.H. [33] explored the trending metaverse technology and found that it covers internet technologies, VR, and AR among other domains that are geared toward a hyperconnected intelligent interface and participation. However, the researchers found that existing metaverse applications are facing an inherent disconnection between the virtual and physical components and interfaces. Consequently, that could use the metaverse extended framework to add the seamless integration of interoperable agents between the virtual and physical environments, which would give rise to the initial theory and practice

of the synthesis of predicted virtual and physical intelligent environments, including future design and potential for connected experiences. Therefore, the subject matter of this research highlighted the gap between the integration of the physical and virtual dimensions within the metaverse. It presented the concern of data connectivity between the virtual and physical environments: if the metaverse and the physical space could not communicate with each other, the interaction between the virtual and physical spaces would not be possible to work together (e.g., static objects in the material world and the metaverse would be incompatible, or full VR immersion would cause glitches and block navigation in the physical space), where inconsistent signal "noise" between them could be overwhelming or irrelevant. As a consequence, this could eventually result in intellectual overload for the user in both the physical and virtual domains simultaneously. Hence, this situation could be annoying to users just because the metaverse would be used more and more, thus disconnections caused by incompatibilities could affect critical usability or critical situations. This research presented a design perspective on the metaverse by underscoring the need to create a more cohesive and more concrete approach to the metaverse with the physical world to improve user interaction. The theoretical foundations of the new metaverse, XR, and IoT, and extended metaverse were identified, and ways to facilitate stronger connectivity were discussed. Two early design projects were presented, thus showing stronger connections in this domain, leading to an interactive and embodied metaverse application adapted to physical and virtual reality. The researchers hoped that this work will further enhance the design considerations within the metaverse research and development community.

## 3. Materials and Methods

The acquisition of this new technology is through requirement analysis, then analysis, design, new system architectures, implementation and testing before final tuning.

### 3.1. The Proposed Metaverse Environment

In order to create a virtual world in the form of an empowered metaverse, our research team reviewed the relevant and inspired research [34], which proposed a framework and settings for presenting complex models with AR and VR using workflows generated from the Unity engine in conjunction with VR headsets to create interactive applications for VR and AR environments.

This would also encourage the user to understand the course content through their surroundings. The design environment of the metaverse platform would be easy to implement and could increase the capability. Furthermore, it would be necessary to have three important tiers: three-dimensional acquisition, linking, and deployment. The relationship between the proposed metaverse and the three tiers could be represented by Equation (1).

$$
\begin{aligned}
&Let\ Metaverse_{extensible} = \\
&Acquisition(3D_{objects}, 3D_{scenes})\ ||Linking(3D_{objects}, 3D_{scenes}) \\
&||\ Deployment(3D_{objects}, 3D_{scenes}).
\end{aligned}
\tag{1}
$$

where 〖Metaverse〗_extensible refers to the metaverse with extensible features; 3D Acquisition refers to the process of acquiring three-dimensional characters, objects and scenes; linking refers to the process combining the results of the first step to create a virtual sequence; deployment refers to the process for implementing the virtual sequences. Equation (1) can be transformed into the architecture of the extensible metaverse as shown in Figure 4.

System architecture of Extensible Metaverse

**Figure 4.** Three-tiered architecture of the proposed metaverse.

By dividing the system architecture into three tiers, it would be easy to update and change the system. Hence, this was the origin of the term 'extensible metaverse'.

### 3.2. Three-Dimensional Acquisition Tier

Three-dimensional acquisition is the first tier, which is responsible for creating elements that would be presented in the virtual world. The three-dimensional acquisition tier allows for the acquisition of virtual characters, scenes, and objects. In this research, Blender was used to derive the three-dimensional elements in the virtual world. Blender not only allowed users to run it in the GUI mode, but it also provided an API call in Python that could be implemented in other languages, such as PHP, by calling the Blender/Python API via the system exec command.

Figure 5 shows the structure of the Blender/Python API, which explains how developers could run the Blender programs through Python scripts. Using the Python 2.93 Blender API, creating a scene in Blender would be easy, whether creating a simple scene with planes, cubes, textures, light sources, cameras, and cubes. Blender API could also be used to render from three dimensions to two dimensions from different perspectives [30].

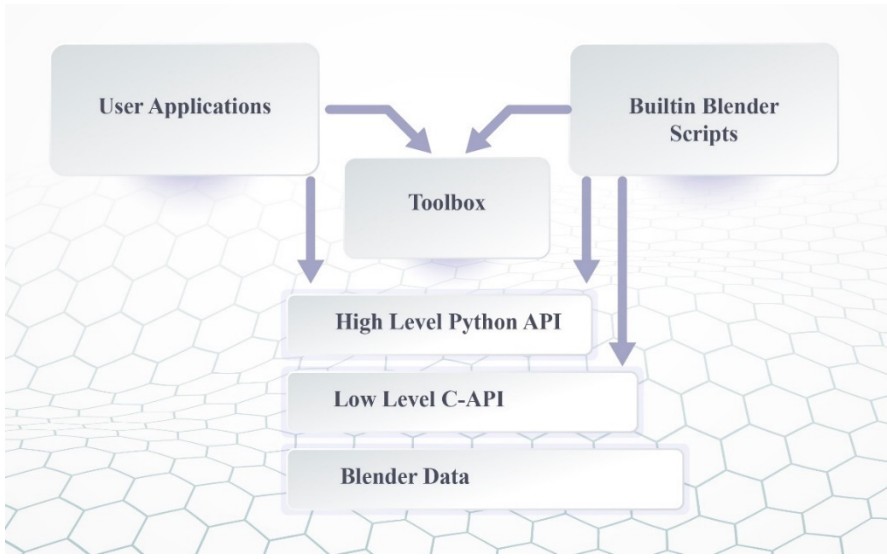

**Figure 5.** Blender/Python API [25].

### 3.3. Linking Tier

This tier combines the characters, objects, and scenes in three dimensions using the Godot Engine, as well as assigning the action to the tourists as characters in the scenes that would enter various activities in each scene of the virtual world by interacting with the scenes, other characters, and virtual objects (Figure 6).

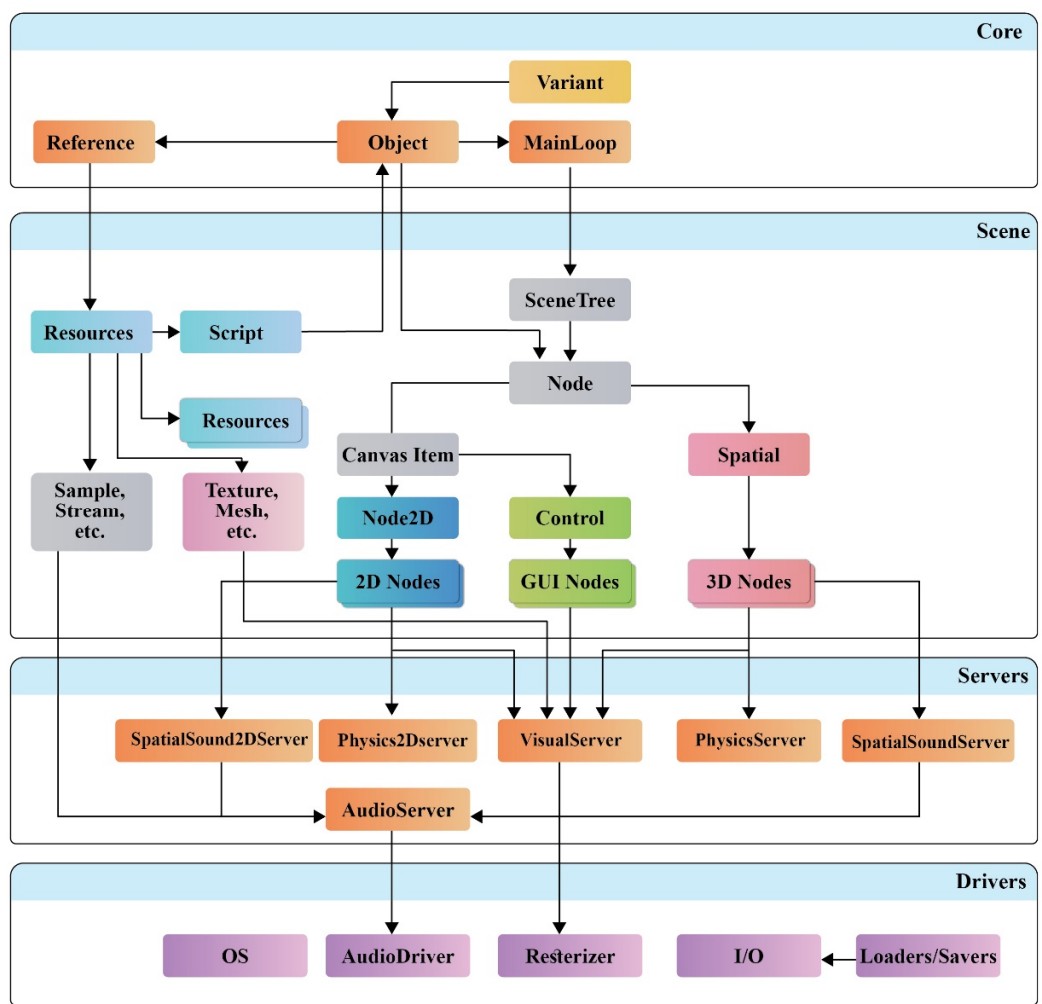

**Figure 6.** Structure of the Godot Engine 3.0 [25].

### 3.4. Deployment Tier

The final tier (Deployment) integrates the results of the Godot Engine with the user database systems and scripting languages, such as PHP, JavaScript, or even Python. These languages and WebGL would be deployed on the machine that has the HTTP server installed.

### 3.5. User Evaluation Metaverse

The research methodology consists of the research design, population, sample, data collection technique, and statistical tools used for the data analysis [35]. In the current study, a quantitative research approach was used, which is also called the deductive approach. The study was conducted to analyze the user's satisfaction of using the metaverse system.

### 3.5.1. Population and Sample

The population comprised domestic tourists in Thailand. The non-probability sampling technique of convenience was used. The sample size was more than 400 participants with a total of 428 based on Cochran (1977) [36] with a confidence level of 95% ($\alpha = 0.05$).

The demographics of the 428 samples found that the majority of tourists were female (332 people or 77.57%) with 96 males (22.43%); most of them were aged between 15 and 24 years (54.44%), earned less than 10,000 THB/month, and were students (43.46%).

### 3.5.2. Data Collection and Analysis

Based on the research conceptual framework and the literature review, the researchers designed a questionnaire with three sections. The first section had five closed questions (gender, age, education, occupation, and income) related to the demographics of the sample. The second section of the questionnaire about evaluating the metaverse consisted of 28 questions (informative, accessibility, interactive, functionality, personalization, and intention to travel). The third section of satisfaction had three closed questions (metaverse satisfaction, tourism experience satisfaction, and overall satisfaction). The questions used a five-point Likert scale ranging as 5 = strongly agree, 4 = agree, 3 = moderate, 2 = disagree, and 1 = strongly disagree. The reliability of the measures was tested using Cronbach's alpha = 0.964. The data were analyzed using SPSS for descriptive statistics. An exploratory factor analysis (EFA) and confirmatory factory analysis (CFA) were run by using LISREL 9.0.

## 4. Results

### 4.1. Implement the Extensible Metaverse in the Research Study

This section shows an example (Figure 7) of how the extensible metaverse would be applied to the tourism industry in Suphan Buri province by adding a section of the data table used to collect the feedback of tourists when visiting each virtual scene in the virtual tourism world.

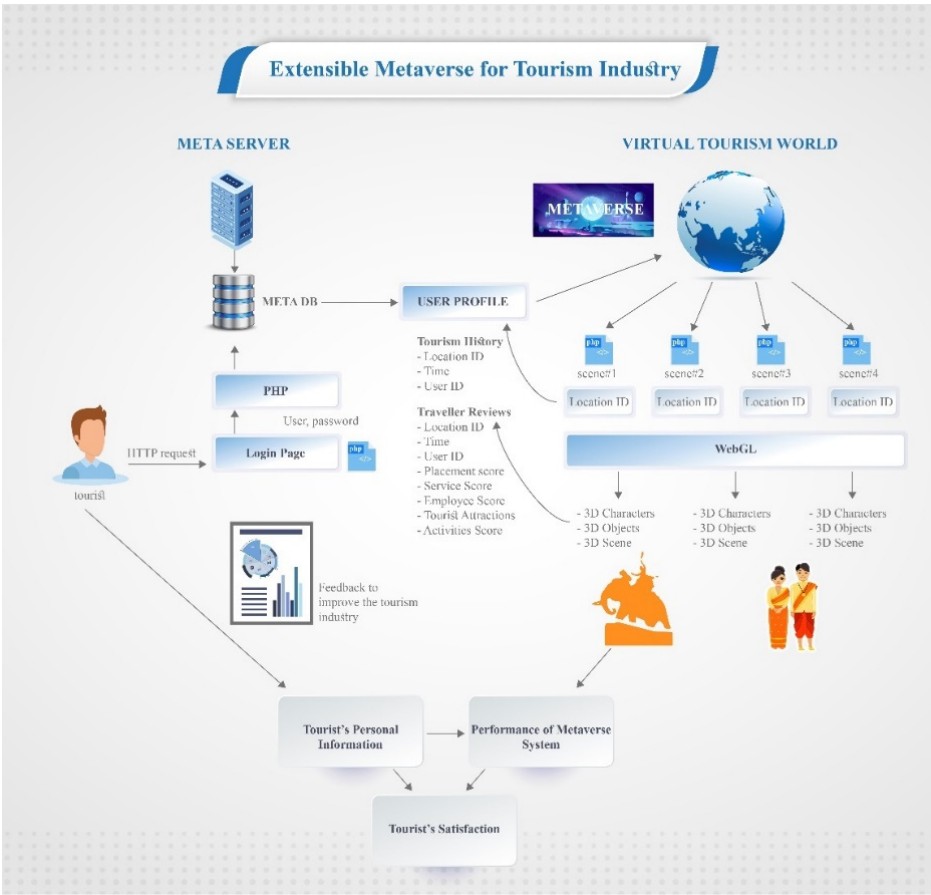

**Figure 7.** Extensive metaverse for the tourism industry.

As a result, this research produced a framework for the metaverse platform that could support the scalability by developers. The developers could freely modify the components and processes in each tier, and when implemented, could apply it to Thailand's tourism industry in Suphan Buri province. It was also found that this customized extensible metaverse worked well through a web browser by working with the popular scripting language used to create the web applications. After the proofing process, the identity users would be able to select the virtual locations from the system's map before going into the three-dimensional details in various formats, as well as being able to rate scenes, objects, or virtual services. As such, travel industry professionals could take these recommendations and opinions into consideration in creating a more satisfying experience for real-world travelers in the future.

After the authentication process, the virtual traveler would enter a map of the virtual world to select the places they wished to visit before deciding on their real-world trip (Figure 8).

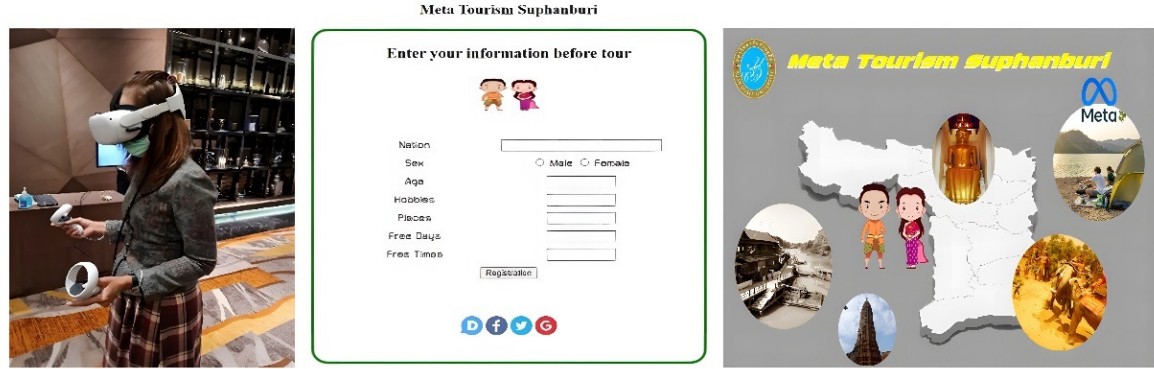

**Figure 8.** Using the extensible metaverse in the Thai tourism industry.

To assess the user satisfaction and perceive the user responses to the Meta Tourism Suphanburi system, the researchers selected usability testing evaluation issues consistent with the research by Gursoya et al. [8] (Figure 9). The traveler assessments were stored in a database and could be retrieved from the SQL Command or through any scripting language in a web browser.

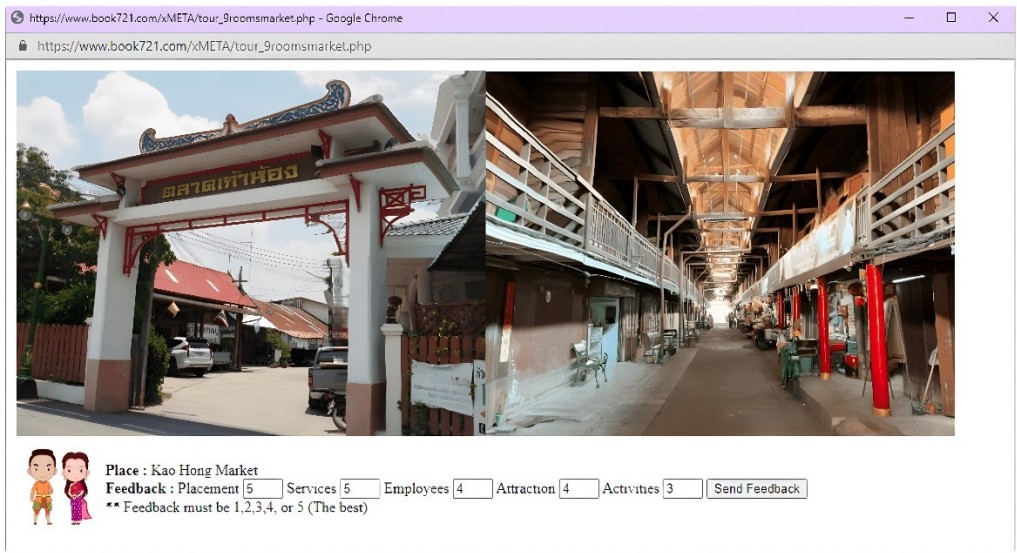

**Figure 9.** Accessing and rating virtual places in Meta Suphan Buri via a web browser.

### 4.2. User Evaluation

The extensible metaverse was implemented in 3 months and evaluated by 428 users. The results found a causal relationship model analysis of tourist behavior toward the use of the extensible metaverse to promote Suphan Buri's tourism. Table 3 describes the correlation coefficient of three latent variables and 13 observable variables. The correlation coefficient was in the range [−0.20–0.595], thus showing the relationship of the observed variables.

**Table 3.** The correlation coefficient.

| Var | Gen | Age | Edu | Occu | Inc | Inf | Acc | Int | Func | Per | Sat1 | Sat2 | Sat3 |
|---|---|---|---|---|---|---|---|---|---|---|---|---|---|
| gen | 1 | | | | | | | | | | | | |
| age | 0.168 ** | 1 | | | | | | | | | | | |
| edu | 0.229 ** | 0.207 ** | 1 | | | | | | | | | | |
| occu | 0.213 ** | 0.509 ** | 0.087 | 1 | | | | | | | | | |
| inc | 0.186 ** | 0.584 ** | 0.506 ** | 0.410 ** | 1 | | | | | | | | |
| inf | −0.061 | −0.028 | 0.060 | 0.038 | 0.024 | 1 | | | | | | | |
| acc | −0.081 | 0.174 ** | 0.103 * | 0.057 | 0.175 ** | 0.531 ** | 1 | | | | | | |
| int | −0.172 ** | 0.010 | −0.013 | −0.070 | 0.037 | 0.439 ** | 0.454 ** | 1 | | | | | |
| func | −0.139 ** | −0.079 | −0.012 | −0.167 ** | 0.031 | 0.595 ** | 0.452 ** | 0.496 ** | 1 | | | | |
| per | −0.147 ** | 0.064 | 0.056 | 0.033 | 0.143 ** | 0.481 ** | 0.408 ** | 0.492 ** | 0.357 ** | 1 | | | |
| sat1 | −0.141 ** | −0.104 * | −0.200 ** | −0.002 | −0.057 | 0.585 ** | 0.424 ** | 0.356 ** | 0.306 ** | 0.472 ** | 1 | | |
| sat2 | −0.150 ** | −0.183 ** | −0.183 ** | −0.143 ** | −0.128 ** | 0.407 ** | 0.452 ** | 0.436 ** | 0.411 ** | 0.488 ** | 0.384 ** | 1 | |
| sat3 | −0.003 | −0.069 | −0.178 ** | −0.104 * | −0.003 | 0.560 ** | 0.399 ** | 0.477 ** | 0.305 ** | 0.348 ** | 0.379 ** | 0.357 ** | 1 |

* The correlation was significant at the level of 0.05; ** the correlation was significant at the level of 0.01.

Figure 10 shows the results of the causal relationship model analysis of tourist behavior toward using the extensible metaverse in Suphan Buri province. This demonstrated that personal had a positive direct influence on the behavior of using the metaverse with an influence size of 0.384.

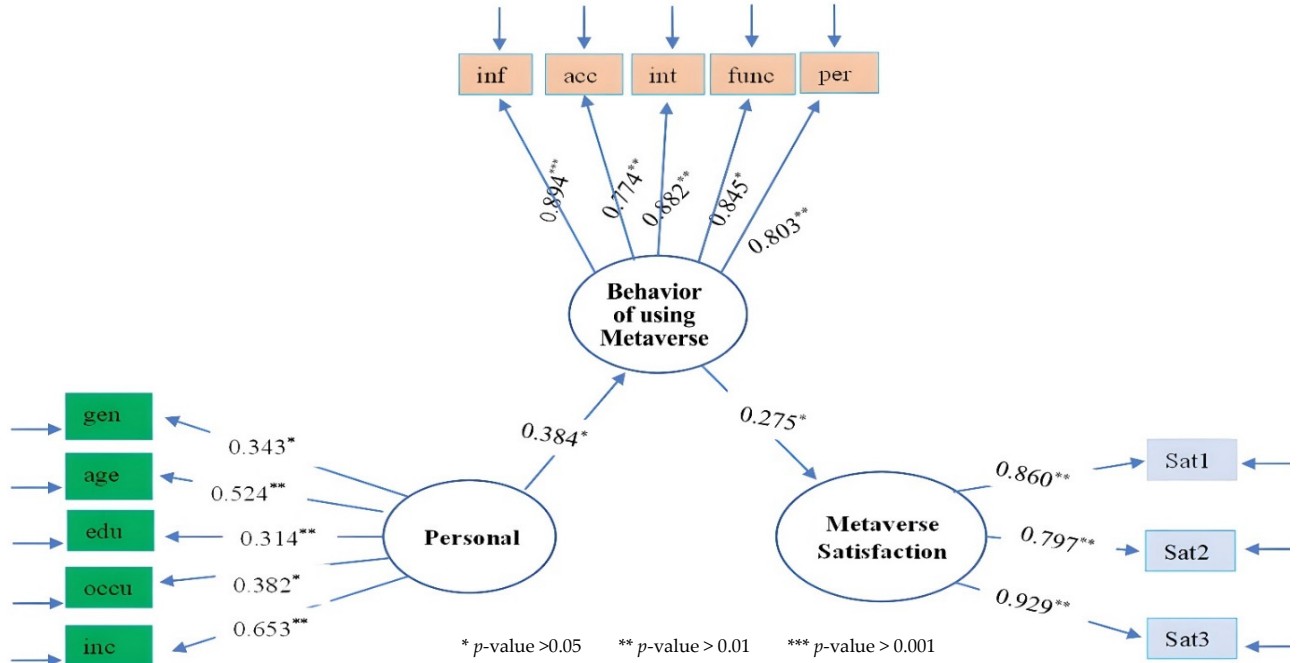

**Figure 10.** The causal relationship model analysis of the user's satisfaction of using the metaverse.

Table 4 illustrates the results of the latent variable analysis. The results found a positive indirect effect of personal and behavior of using the metaverse that had the influence size of 0.106. Finally, the behavior of using the metaverse had a direct effect of satisfaction with an influence size of 0.275. Moreover, Table 5 illustrate the result of the harmonization of the research model with the empirical data (model fit).

**Table 4.** Latent variables of the total effects (TE), direct effects (DE), and indirect effects (IE).

| Latent Variable | Personal | | | Behavior of Using the Metaverse | | |
|---|---|---|---|---|---|---|
| | TE | DE | IE | TE | DE | IE |
| Behavior of using the metaverse | 0.384 * | 0.384 * | - | - | - | - |
| Satisfaction | 0.106 | - | 0.106 | - | 0.275 | - |

* *p*-value < 0.05 (total effects: TE); (direct effects: DE); (indirect effects: IE).

**Table 5.** Harmonization of the research model with the empirical data (model fit).

| Variable | *b* | *SE* | *t* | $R^2$ |
|---|---|---|---|---|
| gen | 0.343 | - | - | 0.3130 |
| age | 0.524 | 0.113 | 14.829 | 0.7820 |
| edu | 0.314 | 0.015 | 9.617 | 0.2380 |
| occu | 0.382 | 0.012 | 14.637 | 0.4760 |
| inc | 0.553 | 0.075 | 10.534 | 0.5860 |
| inf | 0.894 | - | - | 0.7560 |
| acc | 0.774 | 0.062 | 20.468 | 0.5050 |
| int | 0.882 | 0.064 | 19.157 | 0.6560 |
| func | 0.845 | 0.067 | 18.11 | 0.6280 |
| per | 0.803 | 0.063 | 21.379 | 0.5820 |
| Sat1 | 0.860 | - | - | 0.6860 |
| Sat2 | 0.797 | 0.04 | 20.424 | 0.5210 |
| Sat3 | 0.929 | 0.038 | 26.308 | 0.7010 |

$\chi^2 = 47.606$, d = 38, *p*-value = 0.137, GFI = 0.984, AGFI = 0.961
CFI = 0.998, NFI = 0.988, RMR = 0.010, RMSEA = 0.024

The model conformance index validation results pointed out that /df. = 1.253 (less than 2) *p*-value = 0.137, goodness-of-fit index (GFI) = 0.984, adjusted goodness-of-fit index (AGFI) = 0.961, comparative fit index (CFI) = 0.998, normed fit index (NFI) = 0.988 (more than 0.95), root mean square residual (RMR) = 0.010, and root mean square error approximation (RMSEA) = 0.024 (less than 0.05). All indices were appropriate [36]. The details are shown in Table 6.

**Table 6.** Examination of the model conformity index.

| Good of Fit Test | Criteria | Value | Result |
|---|---|---|---|
| $\chi^2$/df | >2.00 | 1.253 | Pass |
| *p*-value | <0.05 | 0.137 | Pass |
| GFI | <0.95 | 0.984 | Pass |
| AGFI | <0.95 | 0.961 | Pass |
| CFI | <0.95 | 0.998 | Pass |
| NFI | <0.95 | 0.988 | Pass |
| RMR | <0.05 | 0.010 | Pass |
| RMSEA | <0.05 | 0.024 | Pass |

Furthermore, the result of user satisfaction with the metaverse system will be helpful in the tourism business and the village to rebuild and encourage tourists to come back post-COVID-19. Moreover, the result of tourists' behavior of using metaverse will be helpful in the business sector and community-based tourism (CBT) to improve the tourism attractions that suit their needs, which will lead to supporting the suitability of tourism and achieving the Sustainable Development Goals (SDGs) model for improving the rural people's quality of life.

## 5. Conclusions and Discussion

In this paper, a new virtual world framework or metaverse, called the extensible metaverse, was presented, which is a three-dimensional virtual world technology that is more flexible than other platforms. Other closed metaverse platforms are difficult for developers to produce and to use across platforms. The researchers also presented guidelines on how to apply the newly developed extensible metaverse to the tourism industry in Thailand, as well as utilizing the feedback of tourists using this new platform to prepare and manage tourism in the post-COVID-19 era in Suphan Buri province. In this regard, the researchers adopted the concept of a smart travel destination (STD) as a new approach that uses innovative information technology to support the hospitality and destination tourism industries, which is consistent with the concept [3,16,17].

The virtual journey in the digital world of the web to listen to audio, read text, and see images of our bodies and the world around us includes unrealistic trips created by the design in the Godot Engine program in this research. This could lead tourists to natural attractions, ancient sites, museums, etc., for users who wish to visit these sites but lack the opportunity to actually visit them; this is consistent with the concepts of Hassani, and Bastenegar [20], Juelu, and Tingting [21], and Guo and Wang [22]. Alternatively, the researchers' vision of the incomplete portion of the original metaverse platform is consistent with the concept of Kshetri [37].

In addition, in this study analyzed the demographics of the 428 samples found that the majority of tourists were female (332 people or 77.57%) with 96 males (22.43%), most of them were aged between 15 and 24 years (54.44%), earned less than 10,000 THB/month, and were students (43.46%). Moreover, we studied the relationship between the use of metaverse technology and tourism in Suphan Buri province through 3 latent variables and 13 observable variables. The results of the causal relationship model analysis revealed that the personal qualities of virtual tourists influenced the use of the metaverse to promote tourism in Suphan Buri province by individual factors in terms of age, income, and occupation. These had correlations of 78.20%, 58.60%, and 47.60%, respectively with the response of metaverse technology and the modern tourism industry, respectively. This information would be very useful for choosing a group of tourists to make public relations and inviting them to travel to Suphan Buri province in the future [2,3,6,19]. Moreover, the Suphan Buri government officers of tourism could implement the extended metaverse tourism system in 30 local communities and train the people from the villages to develop their own content, as well as monitoring the system to support tourism in the post COVID-19 era. This would significantly contribute to the tourism innovation to enhance and rebuild resilience and encourage tourists back to community-based tourism (CBT) by gaining a competitive advantage and achieving the Sustainable Development Goals (SDGs) model for improving the rural people's quality of life [11,19].

For a future study, the research team plans to develop the extensible metaverse platform's capabilities to fully support blockchain, as well as being able to manage digital assets, such as NFT, effectively on technologies related to Web 3.0. This would be consistent with the concept of Kshetri [38–41]. Finally, a future study should focus on international tourists that want to travel to Suphan Buri province and compare the user's satisfaction between the domestic and international tourists' perspective of using the extended metaverse [42,43].

**Author Contributions:** Conceptualization, P.S. and P.P.; methodology, P.S. and P.P.; software, P.S. and P.P.; validation, P.S. and P.P.; formal analysis P.S. and P.P.; investigation, P.S. and P.P. resources, P.S. and P.P. data curation, P.S. and P.P.; writing—original draft preparation, P.S. and P.P.; writing—review and editing, P.S., T.N. and P.P.; visualization, P.S., T.N. and P.P.; writing—review and editing, P.S., C.N., T.C., T.N. and P.P.; supervision, K.J.; project administration, P.S.; funding acquisition, P.S. All authors have read and agreed to the published version of the manuscript.

**Funding:** This research was funded by Suan Dusit University under the Ministry of Higher Education, Science, Research and Innovation, Thailand. The research grant number 65-FF-003 "Innovation of Smart Tourism to Promote Tourism in Suphan Buri Province".

**Institutional Review Board Statement:** The study was conducted in accordance with the ethical and approved by the Ethics Committee of Suan Dusit University (SDU-RDI-SHS 2022-030, 1 June 2022) for studies involving humans.

**Informed Consent Statement:** Not applicable.

**Data Availability Statement:** Excluded the data availability.

**Acknowledgments:** This work was supported by Suan Dusit University and Srisaket Rajabhat University, Thailand. In addition, the research team would like to thank the Suphan Buri Municipality and the Smart City Suphan Buri Committee for all their cooperation and providing the necessary information for the research.

**Conflicts of Interest:** The authors declare no conflict of interest.

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
