# Peer review of "Extensible Metaverse Implication for a Smart Tourism City"

_sustainability, doi:10.3390/su142114027_

Round 1
Reviewer 1 Report
This paper proposes a virtual world framework, or metaverse called extensible metaverse, composed of three-dimensional virtual world technology that is more flexible than other platforms. In addition, a statistical analysis to assess the feasibility of the implementation was performed.
There are some suggestions that authors must take into consideration to improve the quality of the paper:
· There are some typos in the entire manuscript, which must be corrected.
· The abstract should be rewritten, highlighting the main contributions and emphasizing the motivation of this study.
· English must be improved in the entire document.
· It is important to highlight the contributions and describe why this proposal is innovative.
· What areas of computer science are involved in the work? Please provide a table that describes each area and its impact on the research work.
· How this proposal is sustainable for smart cities? Please provide a detailed description concerning this theme.
· How this approach could be immersed in particular domains such as online education or e-tourism? Please provide examples and explanations.
· The figures in the manuscript are very bad and blurry. Please improve its quality and resolution (300 dpi).
· It is imperative to review the state-of-the-art and cite the most important research works concerning “metaverse” in Section 2. Later, a comparison of these proposals regarding your work should be reported.
· Please, carefully with Equations. Equation 1 is not a mathematical representation; it is only a figure or expression of any idea…
· I suggest rewriting or completing Sections 3.1, 3.2, 3.3, and 3.4 with “valuable” information.
· Some descriptive tables of the statistical analysis and a discussion of the results are needed in the manuscript.
Author Response
- There are some types in the entire manuscript, which must be corrected. The abstract should be rewritten, highlighting the main contributions and emphasizing the motivation of this study.
Answer: We rechecked and rewritten the abstract which emphasize on highlighting the main contributions, methodology, result and conclusion of this study (Line 19-38).
- English must be improved in the entire document.
Answer: We rewritten the manuscript and proof the language by the native speaker.
- It is important to highlight the contributions and describe why this proposal is innovative.
Answer: We explain the highlight of the contributions and describe why this proposal is innovative (Line 19-25, 127-143, 152-157,607-616).
- What areas of computer science are involved in the work? Please provide a table that describes each area and its impact on the research work.
Answer: We provide Table 1: Area of applying information technology and the impact of the research work to explain how information technology and computer science are involved in this work and its impacts on the research work in term of AI, Smart Tourism Destination (STD), Smart City, Metaverse, 3D/ Virtual Reality (Line 152-156)
- How this proposal is sustainable for smart cities? Please provide a detailed description concerning this theme.
Answer: We explain the description of how extended metaverse promote the smart cities to sustainability development (Line 164-174, 646-652)
- How this approach could be immersed in particular domains such as online education or e-tourism? Please provide examples and explanations.
Answer: The research approach could be immersed in the particular domains on e-tourism which focusing virtual tourism (Line 164-186,192-221)
- The figures in the manuscript are very bad and blurry. Please improve its quality and resolution (300 dpi).
Answer: We improve the quality and resolution of every figure to be 300 dpi and attachment the file .JPEG of the picture submit to journal for adding into the manuscript.
- It is imperative to review the state-of-the-art and cite the most important
research works concerning "metaverse" in Section 2. Later, a comparison of these proposals regarding your work should be reported.
Answer: We review the current research about “metaverse” and comparison with “extended metaverse” by summary in Table 2. Comparison of the metaverse and extended metaverse (Line 306-307)
- Please carefully with Equations. Equation 1 is not a mathematical representation, it is only a figure or expression of any idea.
Answer: We already recheck Equations 1 that represent the expression (Line 442-456)
- I suggest rewriting or completing Section 3.1, 3.2, 3.3 and 3.4 with “valuable” information
Answer: We rechecked and delete unnecessary information in Section 3.1, 3.2, 3.3 and 3.4 to be condensed and valuable (Line 433-500)
- Some descriptive tables of the statistical analysis and discussion of the results are needed in the manuscript
Answer: We add the statistical analysis and discussion of the results in the manuscript (Line 511-515, 634-637)
Reviewer 2 Report
This paper focuses on the research of the Extensible Metaverse implication in smart tourism city, which is a good research and combination method, and its work has important practical significance. However, I have some comments and suggestions that I hope will help the author improve the manuscript. My comments and suggestions are as follows.
1.In abstract, there are too many background descriptions, and less research methods and results, there is no obvious conclusion. Generally speaking , the abstract should be concise and mainly includes four parts: purpose, key methods, results and conclusions.
2. In the part 1 Introduction and part 2 Review Literature are described the current research status of the Metaverse. It’s too repetitive and verbose. It is suggested that simplify these two parts.
3. Figure 3 in 2.3 Metaverse In Tourism is not that pretty, and Figure 5 in the 3.2 Three- Dimensional Acquisition Tier is not clear enough. I hope these two pictures can be redrawn.
4. The code in 3.2 Three- Dimensional Acquisition Tier has no practical significance, so it can be deleted or implemented with pseudo code.
5. We know that in the Metaverse, the sharing of user experience data is very critical, but the article does not clearly point out the implementation of this technology. I think it’s could be explain it in this paper.
6. There are few references about the Metaverse, so it is suggested to supplement. And I hope some sentences which in this paper can be modified, the language of the paper is not that fluent to read.
Author Response
- ln abstract, there are too many background descriptions, and less research methods and results, there is no obvious conclusion. Generally speaking , the abstract should be concise and mainly includes four parts: purpose, key methods, results and conclusions.
Answer: We rewrite the abstract to be mor concise and include main four part purpose, key methods, results and conclusions. (Line 18-38)
- In the part 1 Introduction and part 2 Review Literature are described the current research status of the Metaverse. It's too repetitive and verbose. It is suggested that simplify these two parts.
Answer: We simplified and condensed the content in Introduction and Review literature section to be short and explain the contribution of this paper.
- Figure 3 in 2.3 Me/averse In Tourism is not that pretty, and Figure 5 in the 3.2 Three- Dimensional Acquisition Tier is not clear enough. I hope these two pictures can be redrawn.
Answer: We redraw Figure 3 and Figure 5 and resolution the picture to be 300 dpi to make it clear (Line 284-285, 473-474).
- The code in 3.2 Three- Dimensional Acquisition Tier has no practical significance, so it can be deleted or implemented with pseudo code.
Answer: We already delete the pseudo code in section 3.2 Three- Dimensional Acquisition Tier (Line 466-473 )
- We know that in the Metaverse, the sharing of user experience data is very critical, but the article does not clearly point out the implementation of this technology. I think it's could be explain it in this paper.
Answer: We explain the implement the extensible metaverse in the research study for test and use the system. After that user evaluate the system (Line 531-566, 568-606)
- There are few reference about the Metaverses, so it is suggested to supplement. And I hope sentences which in this paper can be modified, the language of this paper is not that fluent to read.
Answer: We add more reference about Metaverse [19],[39-44]. Moreover, we English proof by the native speaker.
Reviewer 3 Report
1. Based on your statement, the usage of blender require high processing power to allow the virtual image to look real, is there any solution in doing so.
2. Based on the user evaluation, due to the application is for tourism, how many of the users are foreigner?
Author Response
- Based on your statement, the usage of blender requires high processing power to allow the virtual image to look real, is there any solution in doing so.
Answer: We improve the quality and resolution of every figure to be 300 dpi and attachment the file .JPEG of the picture submit to journal for adding into the manuscript.
- Based on the user evaluation, due to the application is for tourism, how many of the users are foreigner?
Answer: The metaverse implement and evaluated by domestic tourist in Thailand due to our country not really open for the international tourist yet therefore we recommended in the future study should implement on the international tourist that want to travel to Supan Buri province and comparison the user’s satisfaction between domestic and international tourist perspective of using extended metaverse (Line 646-652,656-659)
Round 2
Reviewer 2 Report
Reviewer Blind Comments to Author:
This paper focuses on the research of the Extensible Metaverse implication in smart tourism city, which is a good research and combination method, and its work has important practical significance. The author's answers have solved some of my previous questions already, while I still have some comments and suggestions that could be help author to improve the manuscript. My comments and suggestions are as follows.
1、Figure 3 in 2.3 Metaverse In Tourism is not that pretty. I hope to beautify the layout of the picture, not only to improve the resolution of the picture, but also to redraw it. As for Figure 6 in 3.3 Linking Tier, the font in the picture is too small. I hope to redraw it;
2、In Conclusion & Discussion, there are too many narratives. Generally speaking, the Conclusion and discussion only need to highlight your contribution and innovation, as well as your outlook for the future. There is not need to be too complicated. I hope it can be simplified.
Author Response
- Figure 3 in 2.3 Metaverse In Tourism is not that pretty. I hope to beautify the
layout of the picture, not only to improve the resolution of the picture, but also to redraw it. As for Figure 6 in 3.3 Linking Tier, the font in the picture is too small. I hope to redraw it.
Answer: We draw the new Figure 3 in 2.3 Metaverse In Tourism Figure in 3.3 Linking Tier and make the font bigger easy to read.
- 2. In Conclusion & Discussion, there are too many narratives. Generally speaking,
the Conclusion end discussion only need to highlight your contribution and innovation, es well as your outlook for the future. There is not need to be too complicated. I hope it can be simplified.
Answer: We verify and delete unnecessary conclusion (Line 619-627) and simplify on the contribution and innovation, es well as outlook for the future.
Reviewer 3 Report
The paper is good to be publish
Author Response
The paper is good to be publish
Answer: Thank you so much.